# A natural tandem array alleviates epigenetic repression of *IPA1* and leads to superior yielding rice

Lin Zhang[1,2,3,*], Hong Yu[4,*], Bin Ma[5], Guifu Liu[4], Jianjun Wang[6], Junmin Wang[6], Rongcun Gao[7], Jinjun Li[7], Jiyun Liu[1], Jing Xu[1], Yingying Zhang[1], Qun Li[1], Xuehui Huang[8], Jianlong Xu[9], Jianming Li[2], Qian Qian[9], Bin Han[8], Zuhua He[1,3] & Jiayang Li[3,4]

Super hybrid rice varieties with ideal plant architecture (IPA) have been critical in enhancing food security worldwide. However, the molecular mechanisms underlying their improved yield remain unclear. Here, we report the identification of a QTL, *qWS8/ipa1-2D*, in the super rice Yongyou12 (YY12) and related varieties. In-depth genetic molecular characterization of *qWS8/ipa1-2D* reveals that this newly identified QTL results from three distal naturally occurring tandem repeats upstream of *IPA1*, a key gene/locus previously shown to shape rice ideal plant architecture and greatly enhance grain yield. The *qWS8/ipa1-2D* locus is associated with reduced DNA methylation and a more open chromatin state at the *IPA1* promoter, thus alleviating the epigenetic repression of *IPA1* mediated by nearby heterochromatin. Our findings reveal that IPA traits can be fine-tuned by manipulating *IPA1* expression and that an optimal *IPA1* expression/dose may lead to an ideal yield, demonstrating a practical approach to efficiently design elite super rice varieties.

[1] National Key Laboratory of Plant Molecular Genetics and National Center of Plant Gene Research, Institute of Plant Physiology & Ecology, Shanghai Institutes for Biological Sciences, Chinese Academy of Sciences, Shanghai 200032, China. [2] Shanghai Center for Plant Stress Biology, Shanghai Institutes for Biological Sciences, Chinese Academy of Sciences, Shanghai 201602, China. [3] University of the Chinese Academy of Sciences, Beijing 100049, China. [4] State Key Laboratory of Plant Genomics and National Center for Plant Gene Research, Institute of Genetics and Developmental Biology, Chinese Academy of Sciences, Beijing 100101, China. [5] School of Life Science and Technology, Shanghai Tech University, Shanghai 201210, China. [6] Institute of Crops and Nuclear Technology Utilization, Zhejiang Academy of Agricultural Sciences, Hangzhou 310021, China. [7] Jiaxing Academy of Agricultural Sciences, Jiaxing, Zhejiang 314016, China. [8] National Center for Gene Research, CAS Center for Excellence of Molecular Plant Sciences, Institute of Plant Physiology & Ecology, Shanghai Institutes for Biological Sciences, Chinese Academy of Sciences, Shanghai 200233, China. [9] Agricultural Genomics Institute, Chinese Academy of Agricultural Sciences, Shenzhen 518120, China. * These authors contributed equally to this work. Correspondence and requests for materials should be addressed to Z.H. (email: zhhe@sibs.ac.cn) or to J.L. (email: jyli@genetics.ac.cn).

G lobal food demand is expected to double by the year 2050 (ref. 1). Rice is a major cereal crop that feeds more than half of the world's population. In the past 50 years, rice yield has increased massively due to the identification of semi-dwarf varieties and the development of hybrid rice[2,3]. However, rice production is stagnating and even collapsing in some regions, including China, threatened by the loss of arable farmland and growing population[4,5]. Therefore, it is urgent to develop new genetic resources and strategies to break the bottleneck and meet the increasing food demand.

The advances in high-quality rice genetics, genomics and molecular markers have accelerated the dissection of key genes for yield improvement. The first leap of rice productivity was attributed to the finding and application of the semi-dwarf locus *sd1*, which led to the Green revolution[6], and *sd1* was identified as a mutation in GA20ox-2 that results in reduced biosynthesis of the hormone giberrellin[7]. To further improve yield potential, the International Rice Research Institute raised the concept of new plant type or ideal plant architecture (IPA) for variety development, characterized by few unproductive tillers, more grains per panicle, and thick and sturdy stems[8]. These traits show continuous phenotypic variation and are controlled by multiple quantitative trait loci (QTLs). Several QTLs related to the IPA definition have been cloned, including *Gn1a*, *Ghd7*, *ipa1/WFP*, *SCM2*, *dep1* and *SPIKE*, all of which show great potential in improving rice yield[9–15]. In particular, *IPA1* encodes the SBP-domain transcription factor OsSPL14. A point mutation in *OsSPL14* relieves its repression by the miRNA OsmiR156 and affects all three IPA characteristics simultaneously[9].

The utilization of hybrid vigour (heterosis) has greatly enhanced rice productivity, however, the underlying mechanisms have not been well-characterized. A global view of heterosis has been proposed by re-sequencing more than 1,000 hybrid varieties, which found that a number of superior alleles underlie the high yield potential of hybrids[16]. This finding suggests that heterosis can be achieved by combining key QTLs through an approach known as 'gene pyramiding'. Indeed, this approach is used for breeding many super rice varieties with high productivity[2]. In particular, the widely commercialized super hybrid Yongyou12 (YY12) achieved a productivity of 14.5 t grains per hectare on average, and set a national record. YY12 and related varieties are intersubspecific hybrids (*indica* × *japonica*) exhibiting the IPA phenotypes, but we did not find the *ipa1* allele in these varieties[17], suggesting a novel mechanism contributing to the increased yield.

Here, we report the identification and characterization of a major QTL, *qWS8/ipa1-2D*, from the original breeding stock of YY12 which explains the major IPA phenotypes. The cloned *qWS8/ipa1-2D* locus reveals a previously unidentified large tandem repeats which attenuate the epigenetic repression of *IPA1* promoter and release *IPA1* expression through chromatin remodelling. Our study demonstrates that IPA traits can be optimized by fine-tuning the expression of *IPA1*, and the novel *qWS8/ipa1-2D* locus will enable molecularly designing new super rice varieties with high yield.

## Results

**Map-based cloning of the *qWS8* locus.** To uncover the new factors for the IPA traits in YY12 and related hybrid varieties that are widely grown in China, we generated an F₂ mapping population by crossing the IPA breeding stock YYP1 with the *japonica* variety Nipponbare (NIP). YYP1 bears extremely strong culms, large panicles and modest tiller numbers compared to NIP (Fig. 1a–c and Supplementary Fig. 1). We identified several QTLs by linking genome-wide molecular markers with the stem traits (Supplementary Table 1). We focused on a major QTL on the

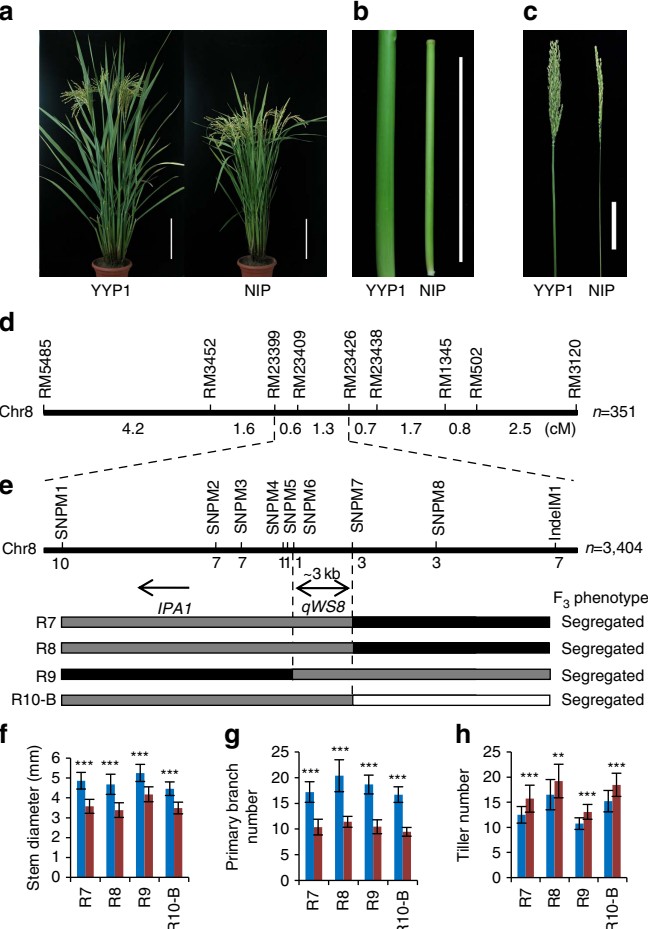

**Figure 1 | Map-based cloning of *qWS8/ipa1-2D*.** (**a**) Morphology of YYP1 and NIP. Scale bars, 20 cm. (**b**) Stems of YYP1 and NIP. Scale bar, 10 cm. (**c**) Panicles of YYP1 and NIP. Scale bar, 10 cm. (**d**) Coarse linkage mapping of *qWS8/ipa1-2D*. (**e**) High resolution mapping of *qWS8/ipa1-2D* to a ~3 kb region of the NIP genome, which is denoted by double headed arrow. Number of recombinants between molecular markers and *qWS8/ipa1-2D* is indicated. The position of *IPA1* is denoted by the single headed arrow. Schematic map of four key recombinants delimiting the mapping region for detailed progeny traits analysis is presented by different bars. Grey bars refer to the heterozygous allele, black and white bars to YYP1 and NIP homozygous alleles, respectively. (**f–h**) Trait comparison of stem diameter (**f**), panicle primary branch number (**g**) and tiller number (**h**) using F₄ sibling lines derived from corresponding recombinants in **e**. Blue and red columns indicate alleles of YYP1 and NIP, respectively. Values are means ± s.d. (n = 24). **P < 0.01 or ***P < 0.001, by Student's *t*-test.

long arm of chromosome 8 that showed the highest logarithm of the odd score and phenotypic contribution, which we named *qWS8* (QTL of wide stem on chromosome 8, also named *ipa1-2D*, see below). To identify the gene underlying *qWS8*, we carried out fine mapping using backcross populations (BC₂F₂), with little interference of other loci (Supplementary Fig. 2). Linkage analysis confirmed the phenotypic effect of *qWS8* (Supplementary Fig. 2), and delineated it to a ~600-kb region (Fig. 1d and Supplementary Fig. 3). A large BC₃F₂ population with 3,404 plants was then screened, and *qWS8* was finally localized between markers SNPM6 and SNPM7 (Fig. 1e and Supplementary Fig. 3). This region covers an ~3-kb interval of the NIP genome with no annotated genes, about 4 kb upstream of *IPA1* (Fig. 1e). Phenotyping of key recombinants indicated that *qWS8* has pleiotropic effects in promoting stem diameter and panicle

primary branch number but reducing tiller number (Fig. 1e–h), similar to *ipa1* (ref. 9). However, more than 10 tillers were generated in homozygous *qWS8* plants, far more than that of *ipa1* and fulfilling the recent demands for IPA rice breeding[18], indicating that *qWS8* only modestly reduces tillering capacity.

We further evaluated the contribution of *qWS8* using an F$_2$ population from YY12 (F$_1$) bearing heterozygous *qWS8* alleles. A significant difference was identified among segregating genotypes and the locus explains as much as 50% of the phenotypic variance for panicle branch and 44% for stem diameter (Supplementary Fig. 4), suggesting that *qWS8* contributes substantially to the IPA traits of YY12. Therefore, cloning of *qWS8* will have a large impact on future super rice breeding.

Next, we investigated the functional sequence variation at this locus. Besides the flanking SNPs (SNPM6 and SNPM7), we did not uncover sequence polymorphisms between YYP1 and NIP by PCR-based sequencing of the mapping region. However, Southern blotting revealed that YYP1 contains a large sequence insertion at the mapping region (Supplementary Fig. 5). Interestingly, digestion with XbaI showed that YYP1 harbours a common band with NIP, as well as a specific ~3-kb band (Supplementary Fig. 5), suggesting a repetitive structure. We then sequenced a BAC containing the *qWS8* locus and discovered that the *qWS8* sequence in YYP1 contains three tandem repeats of the 3,137-bp sequence matching the NIP region, making detection by simple PCR ineffective (Fig. 2a). We therefore postulated that the large tandem repeat region represents the functional polymorphism for the IPA trait determination in YYP1.

***qWS8* origin and allele frequency in natural populations**. To dissect the origin of the tandem repeats in YYP1, we sequenced the whole region covering *IPA1* and *qWS8* from both parents, and found only seven SNPs in addition to the repeats (Fig. 2a). Three SNPs are duplicated within the repeats, suggesting that they appeared before formation of the repeats. Using the seven SNPs as query, we extracted the SNP information from the 3,000 rice-SNP database[19] to find varieties with the same SNPs. We manually excluded varieties with ambiguous and missing information, and obtained SNP types from a total of 2,464 accessions, which form 10 haplotypes (Supplementary Fig. 6). Interestingly, only five *indica* varieties from China share the SNPs identical to YYP1, suggesting that *qWS8* is a rare allele that recently arose in China. Three varieties, GENG77-4, Xiangai B and Jinxibai, were selected for further genotyping by Southern blotting (Fig. 2b). The tandem repeats can only be detected in GENG77-4 and Xiangai B but not in Jinxibai, though they share the same SNPs as YYP1 (Fig. 2b and Supplementary Fig. 6), indicating that the repeats evolved recently from varieties related to Jinxibai during modern breeding. Moreover, Jinxibai exhibits a plant architecture similar to NIP, whereas GENG77-4 and Xiangai B exhibit the IPA phenotype (Fig. 2c–e). This result further supports the notion that the tandem repeats underlie the variation for *qWS8*-mediated IPA traits.

To facilitate genotyping and breeding application, we designed PCR primers spanning the repeat boundary which amplify a 432-bp fragment only from YYP1 but not NIP (Fig. 2a,b). We then genotyped a set of accessions showing IPA traits collected by breeders using a PCR-based approach. Interestingly, we found that more than half of the collections contain the *qWS8* tandem repeats, and share the same SNPs as YYP1 (Supplementary Table 2), suggesting a single origin of this allele. The IPA traits were evaluated and all the varieties with the *qWS8* repeats showed wide stem and high panicle primary branch number (Supplementary Fig. 7). Therefore, the *qWS8* allele has been enriched and adopted widely in recent rice breeding programs.

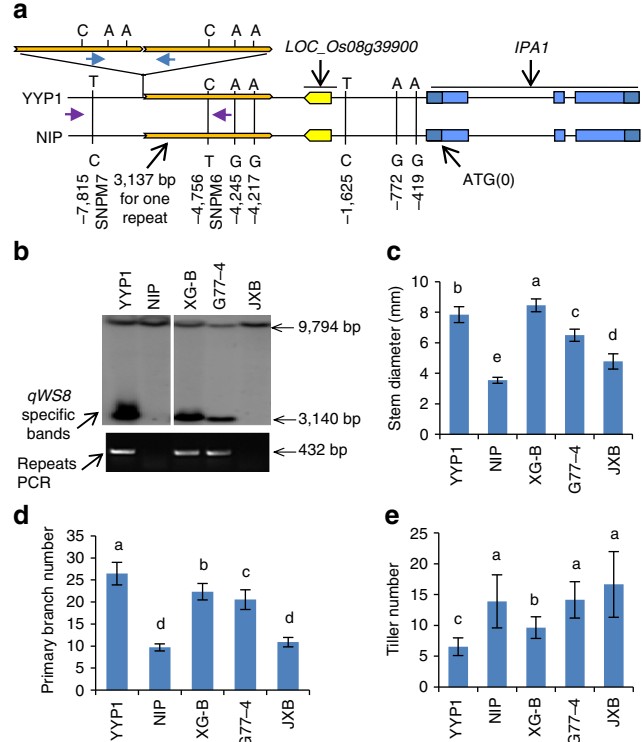

**Figure 2 | Identification of large tandem repeats underlying *qWS8/ipa1-2D* for the IPA traits. (a)** Sequence polymorphisms of the region covering *qWS8/ipa1-2D* and *IPA1* between YYP1 and NIP. Positions of all the SNPs identified were labelled as minus number from ATG (0 position) of *IPA1*. YYP1 bears three copies of the exact NIP sequence (3,137 bp in length) except for three SNPs. The primers only amplifying the repeats were labelled with blue arrows, and primers fail to detect the repeat structure were labelled with purple arrows. **(b)** Genotyping of the three varieties with identical YYP1 SNPs by Southern blotting with XbaI (top) and repeat-specific PCR (bottom) using primers indicated in **a**. XG-B, Xiangai B; G77-4, GENG77-4; JXB, Jinxibai. Note that JXB does not contain the triple repeats. **(c–e)** Phenotypic evaluation of stem diameter **(c)**, panicle primary branch number **(d)** and tiller number **(e)** of five varieties including YYP1 and NIP. Values are means ± s.d. (n = 20). Different letters at top of each column indicate a significant difference at P < 0.05 determined by Tukey's HSD test.

The findings also suggest that the *qWS8*-mediated IPA traits could be stably integrated into different genetic backgrounds.

***qWS8/ipa1-2D* controls IM size by upregulating *IPA1***. Given the proximity of the *qWS8* non-coding region to *IPA1*, we examined *IPA1* expression in the progeny of heterozygous recombinants. We detected higher expression of *IPA1* in plants with the YYP1 allele of *qWS8* compared to plants with the NIP allele (Supplementary Fig. 8). Therefore, *qWS8* is a novel regulatory locus of *IPA1*, which we then renamed *ipa1-2D*. We also renamed the previous *ipa1* allele[9] that abrogates its miRNA-mediated repression as *ipa1-1D* hereafter to uniform the nomenclature. We then developed high-quality near-isogenic lines (NILs) carrying an ~80-kb segment with *ipa1-2D* and *IPA1* (NIP allele), and named them NIL$^{ipa1-2D}$ and NIL$^{IPA1}$ respectively.

We performed qRT-PCR to clarify the tissue-specific expression pattern of *IPA1* and of two miRNAs thought to target *IPA1* at different stages, miR156 and miR529 (ref. 20). *IPA1* was expressed mainly in the inflorescence meristem (IM), and was hardly detected in the seedling, opposite to the pattern of the

two miRNAs (Fig. 3a–c) and consistent with the miRNAs determining the tissue-specific expression of *IPA1*. Further, we found that *IPA1* expression in the IM was upregulated ~3–4 fold in NIL$^{ipa1-2D}$ compared to NIL$^{IPA1}$ (Fig. 3a–c), and obtained similar results by RNA-seq and Northern blotting (Supplementary Fig. 9). These results indicate that the expression of *IPA1* is controlled by both miRNAs and *ipa1-2D*, and that *ipa1-2D* employs a different mechanism to elevate the *IPA1* expression. As *IPA1* is expressed mainly in the IM, we compared its effect on IM development in NIL$^{ipa1-2D}$ and NIL$^{IPA1}$. We found that the IM of NIL$^{ipa1-2D}$ bears a larger size at early stages, and then formed more IM primary branch primordial than that of NIL$^{IPA1}$ (Fig. 3d–i and Supplementary Fig. 10). Therefore, we concluded that *ipa1-2D* determines the IPA phenotypes through tissue-specific up-regulation of *IPA1*. Increased *IPA1* levels promote IM development in rice, laying the cellular foundation for strong culms and large panicles at the mature stage. A similar correlation between large IM size and subsequent branch increase with high productivity has been observed in maize and tomato[21,22].

### *ipa1-2D* alleviates position-dependent chromatin repression.

To explore the molecular mechanism of *ipa1-2D*-mediated regulation of *IPA1*, we first transformed the three tandem repeats into NIP, and analysed the phenotype of T$_1$ generation plants. No obvious morphological difference was found between the transgenic and control plants (Supplementary Fig. 11), suggesting

that the regulation of *IPA1* by *ipa1-2D* may rely on the chromosomal position and act in *cis*. We further generated transgenic plants containing one or three repeats constructs fused to the CaMV35S mini promoter-glucuronidase (GUS) reporter and found that the fusion reporters did not show different GUS expression (Supplementary Fig. 12), suggesting that the tandem repeats do not have enhancer activity.

Genome annotation of NIP revealed that the upstream region of *IPA1* contains many repeats and transposon elements (TEs), which are usually silenced by DNA methylation and form heterochromatin. Histone H3K9m2, a modification associated with heterochromatin, was also enriched in this region (Supplementary Fig. 13). It has been suggested that hetero-chromatin repression can spread to nearby genes[23,24], suggesting a possible mechanism underlying *ipa1-2D*-mediated regulation of *IPA1*. To test this hypothesis, we first examined the methylation status of the region between the *ipa1-2D* repeats and the *IPA1* coding region in the NILs by methylation sensitive Southern blotting (Fig. 4a,b). We found some sites are highly resistant to the digestion, coinciding with the character of hypermethylation in heterochromatin. Nevertheless, we found the region in NIL$^{ipa1-2D}$ was more sensitive to the digestion of different methyl-sensitive restriction enzymes (Fig. 4b and Supplementary Fig. 14). The blotting assays suggest that the reduction in DNA methylation might happen on specific sites upstream of the *IPA1* gene, and the hypomethylation might associate *IPA1* up-regulation with *ipa1-2D* function.

To further clarify the relationship between DNA methylation and *IPA1* regulation, we then performed bisulfite sequencing of the ~800-bp promoter region upstream of the *IPA1* ATG, and found an uneven distribution of three cytosine contexts, with CG and CHG enriched distally, and CHH enriched proximally, which sharply distinguishes two regions of cytosine methylation (Fig. 4c). Importantly, we found a remarkable loss of cytosine methylation in the junction of two regions of NIL$^{ipa1-2D}$ compared to NIL$^{IPA1}$. The CHH was particularly hypomethylated in NIL$^{ipa1-2D}$ compared to NIL$^{IPA1}$ (Fig. 4d–f), suggesting a role for CHH demethylation in *ipa1-2D*-mediated up-regulation of *IPA1*. This is consistent with the recent notion that the CHH islands in the promoter coordinate gene expression near heterochromatin[25].

More importantly, we found that this region contains a DNase I hypersensitive (DH) site (Fig. 4c) and several potentially important *cis*-elements (Supplementary Fig. 15) that could function as binding sites for transcription factors[26]. To test if the large repeats altered the chromatin organization of this region, we evaluated nucleosome organization by a mononuclease (MNase) digestion-PCR approach (Fig. 4g). We found that the region between the repeats and *IPA1* is more sensitive to the MNase digestion, especially for the *IPA1* promoter region (Fig. 4h,i), in NIL$^{ipa1-2D}$ compared to NIL$^{IPA1}$, indicating that the repeats promote an open chromatin state. This result is consistent with the above notion that *IPA1* is repressed by nearby heterochromatin in NIL$^{IPA1}$. To provide further support that the *IPA1* repression is position-dependent, we transformed an 8-kb genomic DNA (gDNA) construct of *IPA1* with its native promoter and coding region containing an artificial SNP to distinguish it from the native gene (Supplementary Fig. 16). Independent transgenic lines showed a higher transgenic mRNA/gDNA ratio than the endogenous gene, indicating a higher transcriptional activity of the transgene (Supplementary Fig. 16), confirming the position-dependent repression of *IPA1*. Collectively, we conclude that the repeat structure in NIL$^{ipa1-2D}$ elevates *IPA1* expression by creating an open chromatin structure which attenuates the epigenetic repression that would otherwise spread from nearby heterochromatin.

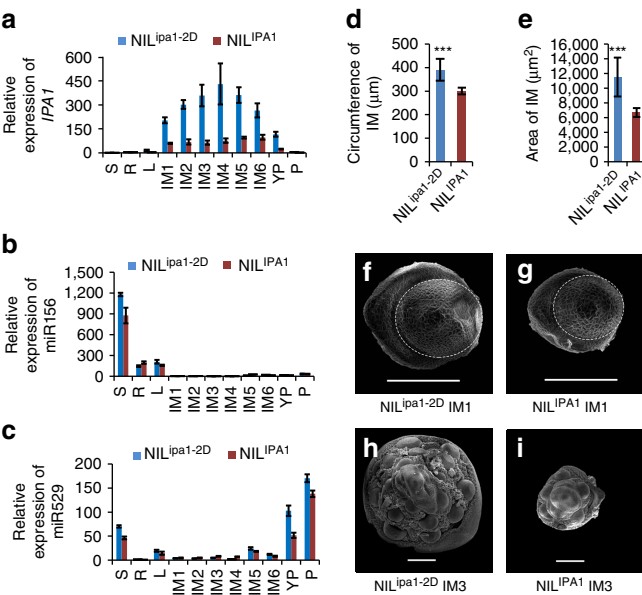

**Figure 3 | Association of *IPA1* and microRNA expression with changes in IM development.** (**a**) Relative expression of *IPA1* in NIL$^{ipa1-2D}$ and NIL$^{IPA1}$, normalized to the rice *Actin* gene, in various organs including seedlings (S), root (R), mature leaf (L), different stages of IM (IM1–IM6), young panicle (YP) and flowering panicle (P). Values are means ± s.d. ($n = 3$). (**b,c**) Relative expression of miRNA156 (**b**) and miRNA529 (**c**) in various organs of NILs, normalized to rice 5 S rRNA. Values are means ± s.d. ($n = 3$). (**d,e**) Comparison of IM circumference (**d**) and area (**e**) between NILs at the IM1 stage in **a**. Values are means ± s.d. ($n = 8$). Triple asterisks represent significant difference between NILs determined by the Student's *t*-test at $P < 0.001$. (**f,g**) Scanning electron microscope images of IMs in NIL$^{ipa1-2D}$ (**f**) and NIL$^{IPA1}$ (**g**) at the IM1 stage in **a**. Scale bars, 50 μm. Dotted lines indicate the regions measured in (**d,e**). (**h,i**) Scanning electron microscope images of IMs in NIL$^{ipa1-2D}$ (**h**) and NIL$^{IPA1}$ (**i**) at the IM3 stage in **a**. Scale bars, 100 μm.

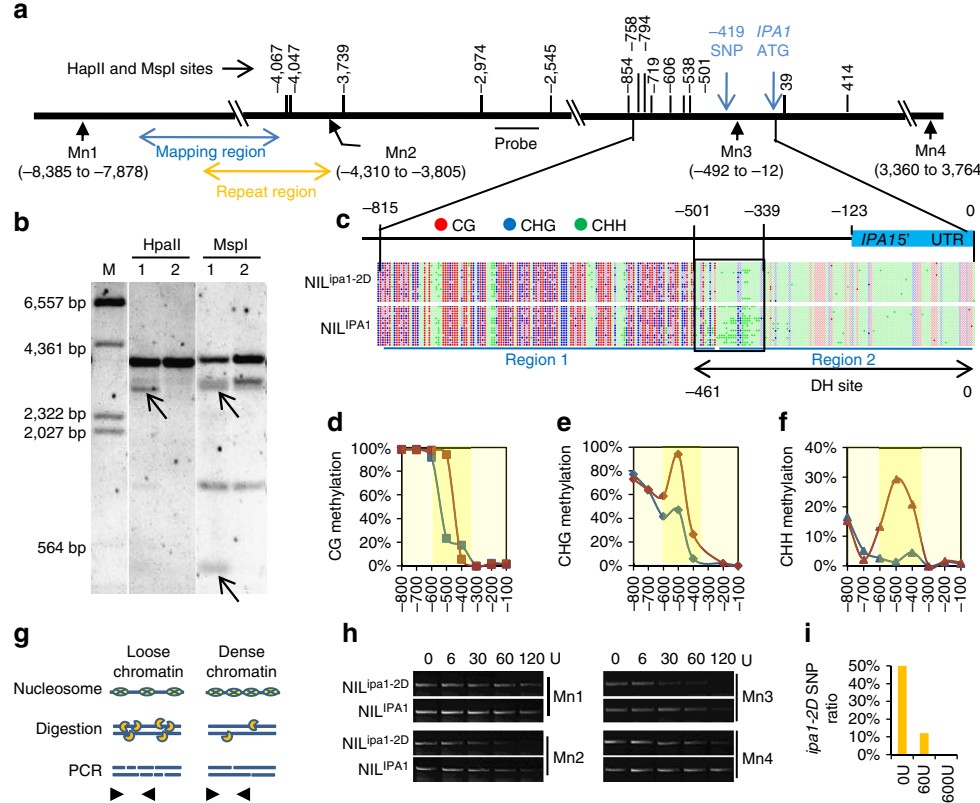

**Figure 4 | Alleviation of epigenetic repression at the *IPA1* promoter by *qWS8/ipa1-2D*.** (**a**) Schematic map showing the sites of two methylation sensitive enzymes (HpaII and MspI) and probe for methylation detection by Southern blotting. The blue and orange double arrows indicate the mapping region and single repeat region respectively. Mn1–Mn4, positions of PCR detection after MNase digestion. (**b**) Southern blot analysis of DNA methylation differences between NIL^ipa1-2D and NIL^IPA1. The bands reflecting different methylation pattern are denoted by arrows. Note that the smallest band is a direct reflection of different methylation at sites −2,545 and −2,974. M, DNA markers. (**c**) Distribution of three cytosine contexts and methylation pattern in ∼800-bp promoter region of *IPA1*. Filled circles, methylated cytosine; empty circles, unmethylated cytosine. Two distinct regions are labelled by blue lines and the junction region is labelled by black frame. Note that the junction region presents obvious methylation difference especially for CHH and overlaps with the DH site labelled by black double arrows. (**d**–**f**) Methylation levels of CG (**d**), CHG (**e**) and CHH (**f**) in eight 100 bp-windows of the promoter region shown in **c**. Blue, NIL^ipa1-2D; Red, NIL^IPA1. Dark yellow shades highlight the junction region. (**g**) Schematic map of the approach to detect open chromatin. Chromatin with loose nucleosome occupation is more susceptible to mononuclease (MNase) digestion and inefficiently amplified by PCR. (**h**) Sensitivity of four regions (Mn1–Mn4) to increasing dosage of MNase digestion between NIL^IPA1 and NIL^ipa1-2D. The position of each region is labelled in **a**. Note that the Mn3 region was more sensitive to the digestion in NIL^ipa1-2D. (**i**) Ratio of the NIL^ipa1-2D allele in the Mn3 region after MNase digestion of nucleus from heterozygous NIL. Two alleles are distinguished by the SNP at position −419 shown in **a**.

**Optimum IPA traits through fine-tuning of *IPA1* expression.** As the new IPA traits of *ipa1-2D* are caused by tissue-specific up-regulation of *IPA1*, we hypothesized that optimal IPA traits can be achieved by manipulating *IPA1* expression dosage. Supporting the hypothesis, varying IPA traits were observed for stable lines with different transgenic copy numbers of two different *IPA1* gDNA constructs (Fig. 5a,b and Supplementary Fig. 17). As copy number increased, the tiller number decreased proportionally, whereas both stem diameter and panicle primary branch number increased (Fig. 5c–e). We analysed the expression level of *IPA1* in more lines and found a close correlation between trait performance and increasing *IPA1* expression (Fig. 5f–h), similar to that with increasing copy number. Moreover, the associations between IPA traits and *IPA1* expression or copy number are non-linear, and the slope of the trait change becomes smooth as *IPA1* expression reaches a plateau (Fig. 5c–h and Supplementary Fig. 17). Together, these results indicate that *IPA1* regulates tiller number, stem diameter and panicle primary branch number in a dosage-dependent manner. It is notable that tiller number was reduced to ∼3 in the plants with the highest *IPA1* expression, resembling the *ipa1-1D/ipa1* phenotypes[9]. Moreover, when *IPA1* was ectopically over-expressed from the

CaMV35S promoter, tillers (average ∼3-4) with strong culms could not be further reduced, suggesting a limit of *IPA1* dosage effect in tillering control (Supplementary Fig. 18). Interestingly, the panicle branch number was not affected in the ectopic over-expression lines, possibly due to the CaMV35S promoter working inefficiently in the process of panicle branch initiation.

Noting that *IPA1* expression has opposite effect on tiller number and panicle branches, two key components controlling yield, we hypothesize that the balance between these traits is a critical factor in shaping the optimal plant architecture for high yield potential. This balance can be obtained by optimizing *IPA1* expression, where sub- and supra-optimal *IPA1* expression will both lower yield potential. Based on this hypothesis, we compared the effect of the naturally occurring *ipa1-2D* allele on yield components under two different growth conditions, Shanghai (eastern China) and Hainan (southern China). Three genotypes with 0, 1 (heterozygous) and 2 (homozygous) copies of the *ipa1-2D* allele were analysed from the NIL population. The IPA phenotypes with larger panicle were observed in plants with *ipa1-2D* allele (Fig. 6a,b), corresponding to higher *IPA1* expression (Fig. 6c). Statistical analysis revealed the large effect of *ipa1-2D* in promoting panicle branch number and spikelet

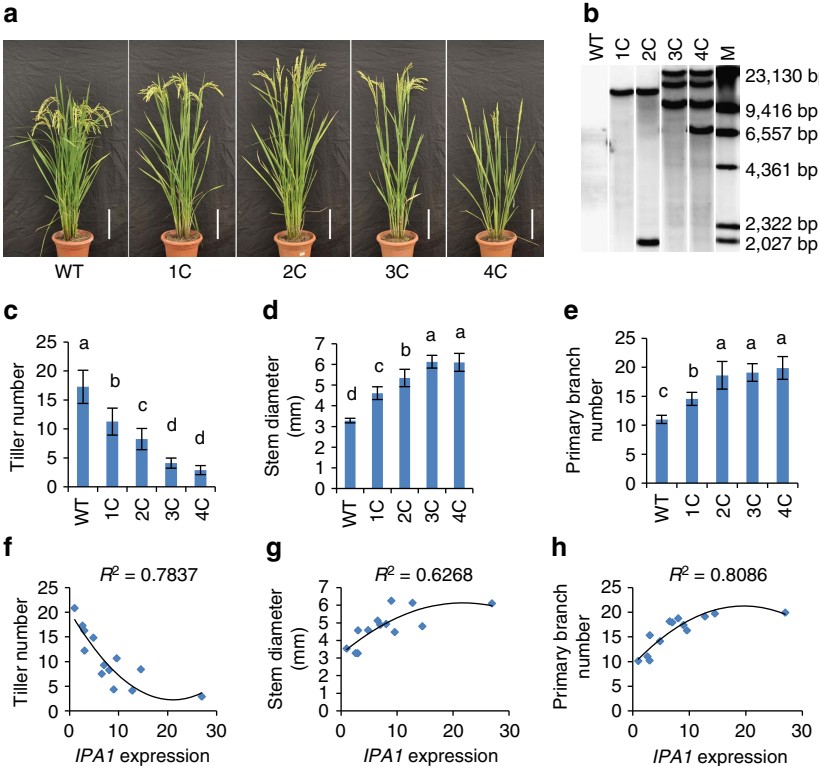

**Figure 5 | *IPA1* dosage affects IPA characteristics.** (**a**) Morphology of plants carrying different *IPA1* gDNA copy number. 1C, 2C, 3C, and 4C represent plants with 1, 2, 3 and 4 insertions of the transgenic construct, compared with the wild-type (WT) NIP. Scale bar, 15 cm. (**b**) Copy number detection of four lines by Southern blotting with WT as control. A probe from hygromycin phosphotransferase gene was used to detect the transgene. M, DNA markers. (**c–e**) IPA traits of plants with increasing *IPA1* copy number, including tiller number (**c**), stem diameter (**d**) and panicle primary branch diameter (**e**). Values are means ± s.d. ($n = 12$). Different letters at top of each column indicate a significant difference at $P < 0.05$ determined by Tukey's HSD test. (**f–h**) Plots of relative *IPA1* expression normalized to rice *Actin* with trait performance in different lines, including tiller number (**f**), stem diameter (**g**) and panicle primary branch number (**h**). Curves fitting the trait change are calculated by quadratic equation with $R^2$ values.

number per panicle, and weak effect in reducing tiller number and grain weight (Fig. 6d–h). Consequently, the yield per plant increased greatly, to as high as 37% in Shanghai and 41% in Hainan (Fig. 6i). The trait variation is consistent at the two stations, indicating the genetic stability of the *ipa1-2D* allele for yield improvement in multiple environments. In addition, NIL^ipa1-2D showed a wider diameter of all internodes and better resistance against both mechanical bending and breaking force than NIL^IPA1, which even did not lodge from strong typhoons in Shanghai in 2013 (Supplementary Fig. 19).

It is worth noting that the yield of the heterozygous plant (*ipa1-2D/IPA1*) inclines toward the level of homozygous plants (*ipa1-2D/ipa1-2D*) and produces mid-parent heterosis of the hybrid of NILs (Supplementary Table 3), in particular in secondary branch number and spikelet number per panicle, suggesting that large panicle produced by *ipa1-2D* is a critical factor both for high yield and heterosis, supporting the notion that the heterozygous *IPA1/ipa1* allele showed strong heterosis in the *indica × japonica* cross[17]. As the yield improvement smoothened in homozygous *ipa1-2D* plants (Fig. 6i), we inferred that this allele could have achieved close to optimal *IPA1* expression, balancing yield components and achieving the superior yield potential. Considering the fruitful effect of *ipa1-2D*, it is rational to expect that the allele will be applied as a primary target for breeding practice.

**Application of *ipa1-1D* and *ipa1-2D* alleles in breeding.** As *ipa1-1D* and *ipa1-2D* had different effects in shaping the IPA traits, especially for tiller number, we clarified their effects on

yield potential in actual breeding. By molecular assisted selection, we *de novo* designed and developed four hybrid varieties. Two of them (JYZK-6 and JYZK-33) bear *ipa1-1D* and another two (JYZK-3 and JYZK-4) bear *ipa1-2D*. Field tests showed that the new varieties, either with *ipa1-1D* or *ipa1-2D*, greatly improved yield performance in comparison with the control variety at two locations of different seasons (as high as 31% in the Zhejiang station and 24% in the Hainan station; Supplementary Table 4), meeting the standard of Chinese super rice. More importantly, the *ipa1-2D* allele could further improve productivity with more tillers in comparison with the *ipa1-1D* allele (Supplementary Table 4). In particular, JYZK-4 showed better yield performance with more productive tillers than the NIL variety JYZK-6, while both of them bear strong culms and large panicle (Fig. 7a,b). Using the $F_2$ population of two hybrids, we accurately determined the contribution of the two alleles in tillering. We found that *ipa1-2D* had a less obvious effect on tillering, whereas *ipa1-1D* caused a significant decrease in tiller number, explaining 34.7% of the total variation (Fig. 7c,d). These findings confirm that *ipa1-2D* confers less of a reduction in tiller number in the varieties.

To compare *IPA1* expression associated with the two alleles in parallel, we obtained plants with *ipa1-1D/ipa1-2D* heterozygous alleles and compared the mRNA levels of two alleles in young panicles (YPs) and seedlings (Fig. 7e,f). The sequencing result showed that the *ipa1-1D* allele was predominantly expressed in both tissues, far more than that of *ipa1-2D* allele (Fig. 7e,f). Similar results were obtained by qRT-PCR analysis of three genotypes in the progeny, and much higher *IPA1* expression was generated in either homozygous or heterozygous *ipa1-1D* plants

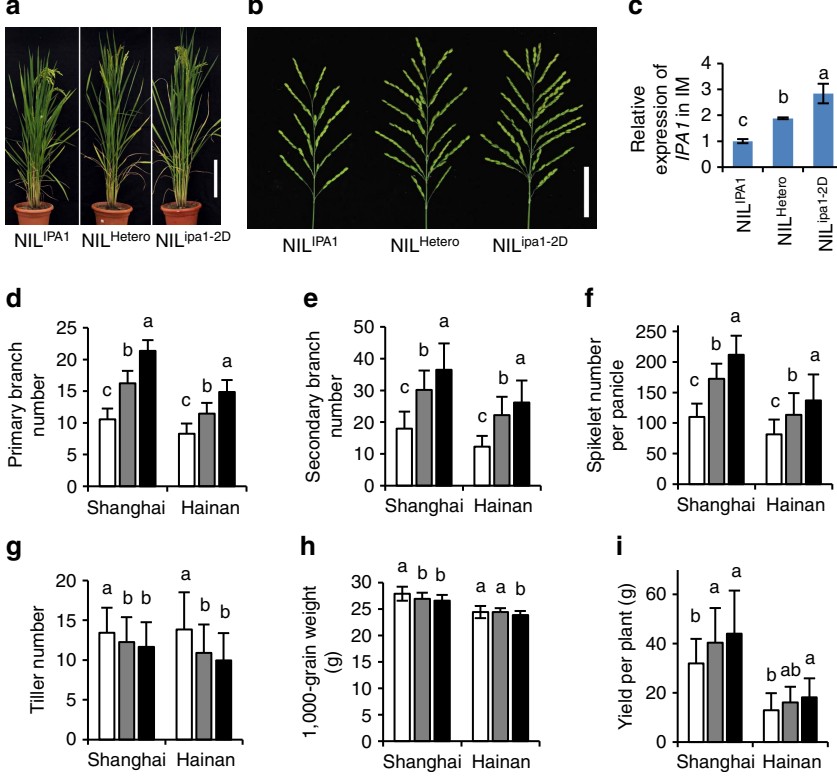

**Figure 6 | Effects of *qWS8/ipa1-2D* allele in reaching optimal yield potential.** (**a**) Overall morphology of three genotypes in the NIL population. Scale bar, 20 cm. (**b**) Panicle morphology of three genotypes. Scale bar, 6 cm. (**c**) Relative expression of *IPA1* in three genotypes, normalized to the rice *Actin* gene. Values are means ± s.d. (*n* = 3). (**d**–**i**) Yield related trait performance of three alleles in the Shanghai and Hainan experimental stations, including panicle primary branch number (**d**), panicle secondary branch number (**e**), spikelet number per panicle (**f**), tiller number (**g**), grain weight (**h**) and yield per plant (**i**). The homozygous *ipa1-2D* and *IPA1* alleles are denoted by black and white columns and heterozygous allele is denoted by grey columns respectively. Values are means ± s.d. (*n* = 67, 124, 62 in Shanghai and 39, 78, 42 in Hainan for three genotypes). Different letters at top of each column indicate a significant difference among genotypes in the respective locations at *P* < 0.05 determined by Tukey's HSD test.

than homozygous *ipa1-2D* plants (Fig. 7g). Together, our data demonstrate that *ipa1-2D* is a naturally occurring weak allele for *IPA1* expression compared to *ipa1-1D*, a naturally occurring strong allele, and *ipa1-2D* generates plants with the optimal combination of tiller number and panicle size to reach the highest yield potential in the tested filed condition and genetic backgrounds (Fig. 7h). Together with *ipa1-1D* and *IPA1* alleles, six different allele combinations will form and certainly provide efficient approaches to generate IPA rice in different environments and genetic backgrounds, fitting diverse cultivation conditions (Fig. 7h). Moreover, pyramiding *ipa1-2D* and *qPL6*, a major QTL for panicle length that we previously identified[27], shaped a better plant architecture with high yield potential (Supplementary Fig. 20). Thus, rice yield potential could be further improved by functionally pyramiding these additive QTLs in future super rice breeding.

## Discussion

Breeding rice with new plant type or IPA to break through the yield ceiling has been proposed since 1990s (ref. 8). However, the molecular mechanisms that generate IPA are not completely understood. In the current study, we reveal that the rice IPA gene *IPA1* is under the control of a non-coding region via a novel epigenetic mechanism. The tandem repeats, a recently occurring regulatory sequence identified in the super hybrid rice YY12 and related varieties, may enhance an open chromatin configuration, and in turn results in a decrease in methylation level of repetitive sequences nested in the proximal promoter of *IPA1* in *ipa1-2D* plants, which is otherwise heavily methylated in the wild-type

(normal) *IPA1* allele. This epigenetic modification may represent a universal mechanism of de-repression for genes proximal to heterochromatin as suggested by recent genome-wide studies[25,28].

Several QTLs underlying important agronomic traits have been mapped to non-coding regions that modify nearby gene transcription[29]. However, the molecular mechanisms of gene transcriptional regulation by these non-coding regions are different, either in *cis* or in *trans*, as indicated by the studies of the maize loci *tb1* and *b1* (refs 30–32). Both *tb1* and *b1* are distant from the functional ORFs (∼60 kb for *tb1* and ∼100 kb for *b1*), but *tb1* confers up-regulation whereas *b1* confers down-regulation of the downstream ORF, respectively. It has been shown that *tb1* covers a functional transposon insertion that up-regulates downstream ORF trascription[31]. In contrast, *b1* was identified as tandem repeats that exert function via trans-acting RNAs, and the phenotype can be regenerated by transforming the repeat sequence only[30,33]. Most recently, we also found the direct copy number variation of the rice grain length gene *GL7* leads to up-regulation of *GL7* and down-regulation of its nearby negative regulator, resulting in an increase in grain length and improvement of grain quality[34]. Interestingly, the three tandem repeats of *ipa1-2D* exert neither trans-acting activity nor enhancer activity, presenting a novel case for non-coding sequence that mediates gene regulation. We provided evidence that the *ipa1-2D* repeats induce an open chromatin status to release the position-dependent repression of *IPA1* in wild-type varieties, probably through an unrecognized chromatin complex. Nevertheless, the mechanisms need to be clarified in depth with advanced techniques in future work.

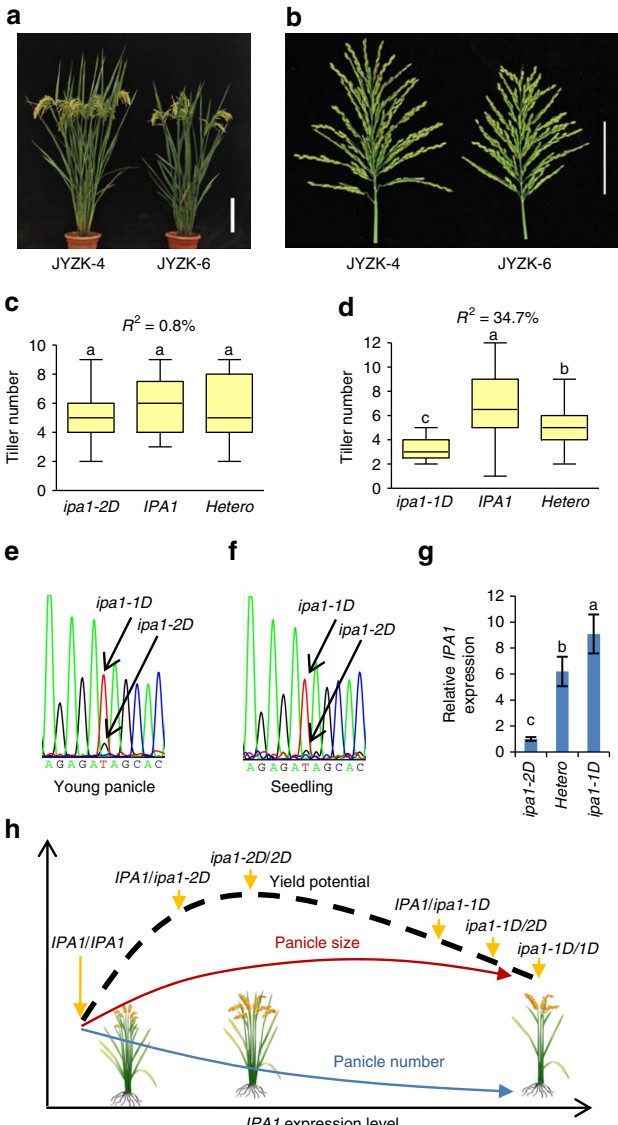

**Figure 7 | *De novo* design of super hybrid rice using *ipa1-2D* and *ipa1-1D*.** (**a**) Overall morphology of newly developed hybrid rice JYZK-4 (*ipa1-2D*) and JYZK-6 (*ipa1-1D*). Scale bar, 20 cm. Note that JYZK-4 has more productive tillers than JYZK-6. (**b**) Panicle morphology of JYZK-4 and JYZK-6. Scale bar, 10 cm. (**c,d**) Contribution of *ipa1-2D* (**c**) and *ipa1-1D* (**d**) alleles to tiller number in the $F_2$ population of the two hybrids as showed by box plot. The centre line, box limits and whiskers represent mean, 25% and 75% confidence limits, and min and max values, respectively. Different letters at top of each box indicate a significant difference among genotypes at $P < 0.05$ determined by Tukey's HSD test. $R^2$ calculated by one-way ANOVA showed the percentage of tiller variation explained by three genotypes in each population, indicating that *ipa1-2D* has a less effect on tillering than *ipa1-1D*. (**e,f**) Chromatogram of *IPA1* transcript abundance of *ipa1-2D/ipa1-1D* plants recovered from RT-PCR with primers flanking the miRNA cleavage site in YP (**e**) and seedling (**f**). Arrows indicate the SNP discriminating the two alleles. (**g**) Relative expression of *IPA1* in three genotypes derived from the *ipa1-2D/ipa1-1D* plant in YP, normalized to rice *Actin*. Values are means ± s.d. ($n = 3$). (**h**) Model of yield potential shaped by the combination of *IPA1*, *ipa1-2D* and *ipa1-1D* alleles. The model predicts that the higher yield can be obtained in plants with modest *IPA1* expression, which generates big panicles and moderate tiller numbers.

The application of rice hybrid vigour or heterosis beginning in the 1970s enabled a substantial yield leap in rice; however, the underlying molecular basis remains largely unknown. Our study indicated that *IPA1* functions as a pleiotropic regulator and determines different IPA traits in a dosage-dependent manner, which provides an alternative mechanistic explanation of heterosis. As *IPA1* expression or copy number increases in transgenic lines, the panicle size increases to a plateau rapidly, whereas the tiller number continues to decrease, indicating that an optimal *IPA1* dosage will confer the best yield potential. Thus, the $F_1$ plants with big panicles and moderate productive tillers can be generated by crossing plants with contrasting *IPA1* dosage, which even forms single-gene heterosis with specific allele combination (Fig. 7h). Coincidentally, dosage sensitivity of the florigen pathway gene also result in yield heterosis in tomato by fine-tuning shoot architecture, and the maximum yield can be generated by combining heterozygous mutations of multiple genes related to floral signal[35,36]. All these results proved the concepts that the dosage effect of gene products is important for the expression of quantitative traits and optimum performance[37]. This is consistent with our recent findings that several heterosis-related genes act through partial dominance for yield components owning to allele dosage and that heterozygous *IPA1* could explain 48.1% of heterosis advantage[17].

IPA has been selected in rice breeding since it was first conceived. However, few super rice varieties with IPA traits have been found containing the *ipa-1D* allele, probably due to its large effect in reducing tillers. In contrast, we found that many super rice varieties carry the *ipa1-2D* allele, indicating that *ipa1-2D* has been preferred by breeders in diverse breeding programs. Compared to *ipa1-1D*, *ipa1-2D* is a weak allele for up-regulation of *IPA1* expression, which confers a large effect on panicle size but little effect on tiller number, resulting in a better overall performance with the increase of grain production of per unit area. Nevertheless, our breeding practice confirmed that *ipa1-1D* could also be a good candidate for hybrid rice development with high yield potential. As higher yield potential could be achieved by integrating the *ipa1-2D* allele with other yield QTLs, we propose that the tiller reduction in both *ipa1-1D* and *ipa1-2D* could be improved by independent high-tillering genes, which will further increase the yield potential. Interestingly, the non-coding repeats of *ipa1-2D* still do not affect the tissue-specific expression of *IPA1*. It raises the possibility to identify novel promoter alleles with better *IPA1* expression pattern to balance panicle size and tiller number either by mutagenesis or genome editing technology. Also, the markers associated with the tandem repeats developed in this study promise to greatly improve rational design breeding in the future.

## Methods

**Plant materials.** YY12 is the well-known commercial hybrid varieties developed and widely grown in China, and YYP1 is the original IPA breeding stock of YY12 with *indica* background. For mapping of *qWS8/ipa1-2D*, YYP1 was crossed with the *japonica* variety NIP, and the resulting $F_1$ plants were selfed to produce $F_2$ generation and were backcrossed with NIP two or three times to generate $BC_2F_1$ and $BC_3F_1$. The $F_2$, $BC_2F_2$ and $BC_3F_2$ populations were used for QTL mapping. A single plant with double recombination covering a ~80-kb heterozygous introgression with *ipa1-2D* was selected from the $BC_3F_3$ generation, and its sibling lines in $BC_3F_7$ generation with homozygous allele of *ipa1-2D* and *IPA1* were selected as NIL$^{ipa1-2D}$ and NIL$^{IPA1}$ by successive selection of single heterozygous plants in each generation followed by a final determination of homozygous alleles. In addition, a $BC_4F_2$ plant with heterozygous *ipa1-2D* and *qPL6* was selfed to generate the gene pyramiding line *ipa1-2D/qPL6* and other combination lines of homozygous alleles.

By searching the SNP database of the 3,000 rice genome[19], GENG77-4, XiangaiB and Jinxibai with the same *qWS8/ipa1-2D* SNPs were ordered and planted for further genotyping and trait evaluation. Since the conceptualization of IPA breeding strategy, rice accessions with varying IPA character has been developed and enriched in China, and we grew 188 such accessions in the Zhejiang

station and Hainan station for genotyping and phenotyping. JYZK-3, JYZK-4, JYZK-6 and JYZK-33 are four hybrids, and their restorer lines were bred by introgressing *ipa1-2D* or *ipa1-1D* alleles from the donor plants assisted by molecular makers. Note that the restorer lines of JYZK-4 and JYZK-6 bear similar background and were crossed with the same sterile lines to facilitate trait comparison.

**Trait measurement.** Plants were grown at paddy field with the same management for agronomic traits comparison. Stem diameter was determined by measuring the third internodes of main culms at mature stage using a slide caliper, and data are presented as means of the major axis and minor axis. Tiller number is counted as all fertile panicles in one plant. Panicle primary branch number, secondary branch number and spikelet number were counted manually from main panicles. Grain yield per plant was measured with all grains from each plant. Maximum bending resistance and breaking resistance were detected by a digital force tester when mature plants were bent to 45° and broken respectively.

For yield test of JYZK hybrid varieties, total grain weight of plants in a plot of $13.34\,m^2$ with three replicates was calculated and normalized to yield per mu ($667\,m^2$). Tiller number was counted as all the fertile panicle in $13.34\,m^2$ region and normalized to number per mu ($667\,m^2$).

**Linkage analysis and QTL mapping.** For QTL detection, 190 plants of $F_2$ population derived from YYP1 and NIP were first genotyped by 145 genome-wide polymorphic markers and linkage analysis was performed by composite interval mapping module of the software WinQTLCart 2.5. Linkage analysis of two $BC_2F_2$ populations with 185 and 166 plants were then performed by single maker analysis module. Progenies of eight $BC_2F_2$ recombinants were planted to clarify the mapping region with valid phenotype. A larger $BC_3F_2$ population with 3,404 plants was genotyped using the markers RM23399 and RM23426 to find additional recombinants, and more markers were developed to discriminate the recombination sites (Supplementary Table 5). Using tightly linked marker, $BC_3F_3$ progeny of key recombinants were genotyped to select $BC_3F_4$ homozygous sibling plants, and traits of both generations were analysed to validate the phenotype. The SNP markers flanking mapping region were also used to genotype different IPA varieties to obtain their SNP haplotypes.

To determine the contribution of *qWS8/ipa1-2D* or *ipa1-1D* alleles in hybrid rice of YY12 or JYZK-4/6, the derived $F_2$ population was genotyped by tightly linked SNP markers, which can distinguish three genotypes in the population. One-way ANOVA analysis was performed to calculate the contribution of allele segregation.

**Southern blot analysis.** Genomic DNA was extracted by the CTAB method and digested by restriction enzymes in $400\,\mu l$ volumes ($10-15\,\mu g$ gDNA, $40\,\mu l$ $10\times$ buffer and $200\,U$ enzymes) for $18\,h$. The digested DNA was extracted using phenol:chloroform treatment followed by ethanol precipitation and transferred onto Hybond $N^+$ membranes (Ameresham) after agarose gel electrophoresis. Probes were synthesized by DIG PCR labelling system and hybridization was performed at $55\,°C$ overnight in DIG Easy Hyb granules (Roche). The membranes were washed in $2\times$ SSC and 0.1% SDS at $65\,°C$ for $15\,min$ twice and then subjected to chemiluminescent detection following the instruction of Detection Starter Kit (Roche). High-quality images were captured by the Tanon-5500 Chemiluminescent Imaging System (Tanon Science & Technology).

**Determination of *IPA1* transcription and microRNAs.** Sequential stages of IM development were obtained by sampling the IM every 2 days starting from the booting stage (time of IM initiation) for RNA preparation. Total RNAs were prepared using a TRIzol kit according to the user's manual (Invitrogen). For detection of *IPA1* transcripts, $1\,\mu g$ of total RNAs was used for cDNA synthesis with a reverse transcription kit (TaKaRa). qRT-PCR was performed in a $20\,\mu l$ volume with $2\,\mu l$ cDNA, $0.5\,\mu M$ gene specific primers (Supplementary Table 5) and $10\,\mu l$ $2\times$ mix (TaKaRa) supplemented to $20\,\mu l$ by water on an ABI 7900 real-time PCR machine according to the manufacturer's instruction (Applied Biosystems). The rice *Actin* gene (*LOC_Os03g50885*) was used as the internal control. Northern blot analysis was carried out to confirm *IPA1* transcript levels with the rice *Actin* probed as loading control. For detection of miR156 and miR529, the same RNA were reverse-transcribed by the miRNA RT-PCR Kit (TaKaRa) and qRT-PCR was performed following the same procedure described above, and universal adaptor primers (supplied by the Kit) and Uni-miRNA primers were used with 5 s rRNA as internal control (Supplementary Table 5).

**Bisulfite sequencing-based DNA methylation analysis.** Genomic DNA from IM tissues was extracted using CTAB method. Bisulfite treatment was performed using the EpiTect Bisulfite kit (Qiagen). Bisulfite-treated DNA was then used to amplify *IPA1* promoter regions from the different genotypes by PCR using primers listed in Supplementary Table 5. Amplified PCR fragments were cloned into pGEM-T easy vector (Promega) and sequenced. Sequences of 17 colonies from each genotype were analysed with Kismeth software[38] to obtain the percentage of methylated sites

for three cytocine contexts. Results were confirmed with more than 3 independent experiments.

***IPA1* transgene expression ratio detection.** For transgene ratio detection, the genomic DNAs and cDNAs from IM tissues of transgenic plants ($T_0$) harbouring the *IPA1* construct with an artificial SNP (see below) were amplified separately using primers in Supplementary Table 5, and cloned into pGEM-T Easy vector, and individual colonies were sequenced to determine the SNP types (transgenic or native *IPA1*). The number of colony with respective SNP type was subjected to the ratio calculation.

**Open chromatin detection by MNase digestion.** Open chromatin was detected using a published protocol[39]. In brief, 2-week-old seedlings of NILs frozen in liquid nitrogen were ground to power and suspended in $10\,ml$ of ice cold nuclei isolation buffer (1 M hexylene glycol, 20 mM PIPES-KOH [pH 7.6], 10 mM $MgCl_2$, 1 mM EGTA, 15 mM NaCl, 0.5 mM spermidine, 0.15 mM spermine, 0.5% Triton X-100, 10 mM β-mercaptoethanol and $1\times$ protease inhibitor cocktail (Roche)) with gentle rotation for $15\,min$. The suspension was filtered through 30-nm CellTrics, and the elute was centrifuged for $10\,min$ at $1,500\,g$ at $4\,°C$. The pellet was resuspended as crude nuclei extract with digestion buffer (40 mM Tris-HCl [pH 7.9], 0.3 M Suc, 10 mM $MgSO_2$, 1 mM $CaCl_2$ and $1\times$ protease inhibitor (Roche)). Equal aliquot of crude nuclei extract was subject to digestion with increasing MNase amounts at $30\,°C$ for $20\,min$ and DNA was extracted using phenol:chloroform treatment followed by ethanol precipitation and used for PCR analysis.

**Constructs for genetic transformation.** The region containing all the three repeats of *qWS8/ipa1-2D* was cloned into binary vector pCambia1301 and transformed into NIP for analysis of possible function of the repeats. The same repeats region was then fused with CaMV35S mini promoter driving the GUS reporter for reporter analysis. The single repeat from NIP was also cloned and fused with the reporter construct. These constructs were transformed in NIP callus to get independent transformants, which were grown in medium supplemented with X-Gluc to detect reporter activity. Three genomic fragments containing *IPA1* with size of 8, 11.9 and 13.9 kb from the NIP BAC AP006049 were cloned into pCambina1301, respectively, and an artificial SNP was introduced into the 8-kb construct by site-directed mutagenesis. The *IPA1* cDNA AK107191 was also cloned into pCambia1301 with the CaMV35S promoter to obtain ectopic *IPA1* over-expression construct. All the gDNA and cDNA constructs of *IPA1* were transformed into NIP to generate more than 15 independent lines. Stable transgenic lines ($T_2$ generation) were used for gene expression and phenotype analysis.

**Inflorescence meristems imaging and measurement.** IM tissues at different stages were sampled and fixed in the FAA solution (45% ethanol, 5% acetic acid, 5% formaldehyde). For scanning electron microscopy, the samples were critical-point dried in liquid $CO_2$ and coated with gold, followed by visualization with a scanning electron microscope (JSM-6360LV, JEOL). For histological sections, samples were embedded in paraffin or resin, and transverse sections were made at $10\,\mu m$ for paraffin samples and $2\,\mu m$ for resin samples. Sections were examined microscopically (BX51, Olympus) and photographed. The IM size was measured from pictures of scanning electron microscopy using ImageJ software (NIH).

**Data availability.** The authors declare that all data supporting the findings of this study are available within the manuscript or its supplementary files or are available from the corresponding authors upon request.

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

## Acknowledgements

We thank Linyou Wang, Lixia Zhang and Yongbin Qi for rice growth and paddy field tests, Zhikang Li for providing rice germplasms, Ertao Wang for valuable suggestions. This work was supported by grants from Ministry of Science and Technology of China (2013ZX08009-003-001, 2016YFD0101800, 2014AA10A600), the Chinese Academy of Sciences (XDA0801 and XDA0803), the National Natural Science Foundation of China (31600990).

## Author contributions

J.L., B.H., Q.Q., Z.H., and J.L. conceived and designed the research. L.Z. and H.Y. designed experiments and analysed the data. L.Z., H.Y., B.M., G.L., J.W., J.W., R.G., J.L., J.L., J.X., Y.Z., Q.L., X.H. and J.X. conducted the experiments. J.L. and Z.H. oversaw the entire study. L.Z., H.Y., Z.H. and J.L. wrote the manuscript.

## Additional information

**Competing interests:** The authors declare no competing financial interests.

