## [Peer Review File · Nature Communications]

Reviewers' comments:

Reviewer #1 (Remarks to the Author):

This manuscript describes the identification of an unusual tandem array that releases the repression of the IPA1 gene in rice. Typically tandem arrays cause silencing but in this case there is a reversal of this role. The results are clear and validated via transgene analysis. The authors go on to illustrate a rheostat on the IPA1 gene would lead to the optimum plant characteristics in breeding. In other words, the amount of gene product and its context determines the optimum results. This is an important concept that is beginning to emerge. It might be worth adding a few citations on the experimental (PloS Genetics 9: e1004043 & Nature Genetics 46: 1337-1342) and theoretical levels (Plant Sci 245: 128-134) to bolster this idea that the effective dosage of gene products and the context in which they operate is important for optimum performance and the expression of quantitative traits.

Minor issues:

Line 340, use semicolon instead of comma.

Lines 350-351 appear to need corrections.

Reviewer #2 (Remarks to the Author):

This paper describes the cloning of a locus associated with changing the plant architecture of rice to lead to higher yields. Several years ago, there was a consensus that increasing yields in rice could accompany changes in plant architecture, which includes among others greater grain number, stiffer stems and lower tiller numbers. Several QTLs have been identified that contribute to this ideal plant architecture (IPA), and in this paper the investigators describe a novel regulatory element that may confer this trait.

The gene IPA1 had been identified that contributes to this desired architecture, and examination of high yielding hybrids have identified a QTL close to this gene that affects this trait. The investigators clone this locus, find that it is close to IPA1, and find it is associated with large tandem repeats. The investigators associate increased number of repeats with increase expression of IPA1, and increased inflorescence meristem size. Based on epigenomic profiling, it appears that increased tandem repeat numbers may alter the epigenetic profile around IPA1 and lead to the phenotype.

This is a well-done paper that goes from gene mapping, cloning, expression and dissecting molecular mechanism, to phenotype, to breeding. It is a very nice interweaving of different links in the chain from gene to breeding response. Although they did some transgenic work that buttresses their claims, it would have been nice to have a definitive transgenic test of the relationship of the locus to phenotype. However, I do not think this is strictly required given how well everything falls into place.

The only real issue I have is they use the word "serial" in several places, to denote what appears to be progeny (I think). It may be best to define this in their context as it is not

usual terminology that is widely known.

Reviewer #3 (Remarks to the Author):

This manuscript describes the genetic mapping and characterization of a locus, qWS8/ipa1-2D, that substantially contributes to traits important for yield enhancement in rice. By itself, this is an important contribution to ongoing efforts to breed and understand loci that are necessary for increasing yield in this agronomically important plant. Although I am not a breeder, this part of the manuscript seems quite well done, and clearly demonstrates that it is a non-coding region upstream of IPA1 that causes differences in expression of this gene. This is in line with many other observations that suggest that regulatory changes are important sources of variation in traits. The authors demonstrate that the genetic change in ipa1-2D results in increased expression of IPA1, and that this expression change is associated with changes in important aspects of plant architecture. Interestingly, although qWS8 was mapped using SNPs, the key change appears to be a tandem duplication several kb upstream of the IPA start of transcription. The authors nicely confirm this by finding that other accessions of rice that have the SNPs but lack the tandem duplication do not exhibit changes in plant architecture. The authors demonstrate that the tandem duplication by itself, when transformed in trans, does not enhance expression of IPA1, nor does it, by itself, act to enhance expression of GUS. Although one could quibble with negative results, they were reasonable tests of the hypotheses. Surprisingly, the authors find evidence for epigenetic differences, or at least differences in DNA methylation, between NILs carrying ipa1-2D and those that lack it. Although fascinating, this is where I begin to have some problems with the analysis. As I will detail below, I had some technical problems with the analysis as presented. In addition however, I have some difficulties with cause effect relationships, primarily because there is no indication presented as to what actually cause the differences in DNA methylation observed here are achieved. However, as discussed below, I suspect that differential methylation observed just upstream of the transcriptional start site of IPA1 are in the edge of a repetitive element. This boundary effect has often been observed in maize, and is associated with differences in the chromatin environment in transposable elements immediately adjacent to genes. In any event, technical issues aside, and aside from a lack of a final understanding of the relationship between DNA methylation, the tandem duplication and increased expression of IPA1, I am cautiously positive about the phenomenology presented here.

Specific comments:

line 78: "reveals a previously unidentified large tandem repeats which attenuates the epigenetic repression of IPA1 through modifying the epigenetic landscape of the IPA1 promoter." Some caution is required here. Certainly, the presence of the tandem repeat is associated with increased expression, which is associated with differences in DNA methylation, but it would also seem possible that the observed changes in DNA methylation are a consequence, not a cause, of increased expression.

line 81: "The superior performance of this novel qWS8/ipa1-2D locus will enable geneticists and breeders to molecularly design new super rice varieties with high yield." perhaps, but if this really is an epigenetic phenomenon, this might be more difficult that the authors

suspect.

line 113: "However, Southern blotting revealed that YYP1 contains a large sequence insertion at the mapping region" I've done quite a bit of restriction mapping, and I have a hard time understanding how the provided Southern blot data demonstrates this, unless a corresponding map of the NIP sequence is provided, along with the expected sizes on that map as well. That being said, since the corresponding region has been sequenced in both accessions, I'm not sure what the Southern data adds. Perhaps it would be better as supplemental data. Further, here and throughout the images have been heavily processed, to the point where the background cannot be seen. Given recent controversies, it would be best if all images were processed as little as possible. In figure 2b, I was a bit confused, since based on my analysis, Os08g39890.1 is IPA1, not adjacent to it. Figure 2c is far too processed, although I accept the results of the analysis.

line 116: "We sequenced a BAC containing the qWS8 locus and discovered that the qWS8 sequence" Did you also sequence the NIP region?

line 140: "We used these primers to genotype a collection of IPA varieties with varying degrees of IPA traits at two growth locations." I'm not sure what is being portrayed here. Are these multiple accessions plotted out?

line 153: "We detected higher expression of IPA1 in plants with the YYP1 allele of qWS8 compared to plants with the NIP allele (Supplementary Fig. 7)." Since this is a quantitative argument, real time PCR should have been used, rather than regular PCR with an unknown number of cycles.

line 197: "We found that the region in NILipa1-2D was more susceptible to digestion with the methyl-sensitive restriction enzymes HpaII and MspI (Fig. 4b). A similar result was obtained with other methylation sensitive enzymes (Supplementary Fig. 13)." This is where I get less comfortable. While I agree that there is some evidence for a decrease in methylation, this is restricted to only a few restriction sites. Further, the blots reveal that the effect is relatively minor, and only one sample of DNA was used for each genotype, and we have no much variation there is between samples. Finally, the XbaI/Alw44I digest is so heavily manipulated that it is hard to know what we are seeing, and there is no indication on the map which sites are differentially methylated. Although I'm tentatively convinced that there is some degree of difference in methylation in this upstream region, this analysis would have to be redone to be convincing. Repetitive sequences can be bisulfite sequences if at least one primer is in a single copy sequence. I would suggest the authors use their methyl sensitive enzyme assays to focus on specific regions that they think are hypomethylated and examine them in more detail.

Line 198: "The blotting assays indicated the existence of highly suppressed chromatin in the IPA1 promoter that could be alleviated by ipa1-2D." No, it simply suggests that some fraction of methylation is lost at a few sites in regions upstream of the IPA1 gene.

line 203: "Importantly, we found a remarkable loss of cytosine methylation in the junction of two regions of NILipa1-2D compared to NILIPA1. The CHH was particularly hypomethylated in NILipa1-2D compared to NILIPA1 (Fig. 4d-f), suggesting a role for CHH demethylation in ipa1-2D-mediated up-regulation of IPA1." This is an interesting result. A quick look suggests to me that the methylated region upstream of IPA1 is repetitive, and likely a transposon. The pattern of methylation at the margins between transposons and genes in grasses are often composed of CHH islands such as the one observed here. This would be worth more consideration. Here and in the previous blotting section, I'd like to

know more about the nature of the sequences being differentially methylated.

line 230: "As copy number increased, the tiller number decreased proportionally, whereas both stem diameter and panicle primary branch number increased" It also looks as if yield is decreasing. Is that true?

line 314-320. I would suggest that these claims are premature. The authors certainly see an association between some differences in methylation, but I don't think they have established that it is causal. It is possible, for instance, that regulatory sequences enhance an open chromatin configuration in *ipa1-2D*, which, in turn results in decreases in methylation of repetitive sequences near the proximal promoters. In this case, changes in methylation would be a consequence, not a cause, of enhanced activity.

Point-by-point response to Reviewers

Response to the comments of Reviewer #1:

This manuscript describes the identification of an unusual tandem array that releases the repression of the IPA1 gene in rice. Typically tandem arrays cause silencing but in this case there is a reversal of this role. The results are clear and validated via transgene analysis. The authors go on to illustrate a rheostat on the IPA1 gene would lead to the optimum plant characteristics in breeding. In other words, the amount of gene product and its context determines the optimum results. This is an important concept that is beginning to emerge. It might be worth adding a few citations on the experimental (PloS Genetics 9: e1004043 & Nature Genetics 46: 1337-1342) and theoretical levels (Plant Sci 245: 128-134) to bolster this idea that the effective dosage of gene products and the context in which they operate is important for optimum performance and the expression of quantitative traits.

Response: Many thanks! As suggested, we have added more discussion of this concept and the references as well.

Minor issues:

Line 340, use semicolon instead of comma.

Response: We have corrected it.

Lines 350-351 appear to need corrections.

Response: Thanks. We have rewritten the sentence.

Response to the comments of Reviewer #2:

This paper describes the cloning of a locus associated with changing the plant architecture of rice to lead to higher yields. Several years ago, there was a consensus that increasing yields in rice could accompany changes in plant architecture, which includes among others greater grain number, stiffer stems and lower tiller numbers. Several QTLs have been identified that contribute to this ideal plant architecture (IPA), and in this paper the investigators describe a novel regulatory element that may confer this trait.

The gene IPA1 had been identified that contributes to this desired architecture, and examination of high yielding hybrids have identified a QTL close to this gene that affects this trait. The investigators clone this locus, find that it is close to IPA1, and find it is associated with large tandem repeats. The investigators associate increased number of repeats with increase expression of IPA1, and increased inflorescence meristem size. Based on epigenomic profiling, it appears that increased tandem repeat numbers may alter the epigenetic profile around IPA1 and lead to the phenotype.

This is a well-done paper that goes from gene mapping, cloning, expression and dissecting molecular mechanism, to phenotype, to breeding. It is a very nice interweaving of different

links in the chain from gene to breeding response. Although they did some transgenic work that buttresses their claims, it would have been nice to have a definitive transgenic test of the relationship of the locus to phenotype. However, I do not think this is strictly required given how well everything falls into place.

Response: Thank you very much for the encouraging comments. We did have tested the relationship between the locus and phenotypes by transforming the three repeats of *ipa1-2D* (Supplemental Fig. 11). We did not find any effect of the tandem repeats alone on the IPA traits.

The only real issue I have is they use the word “serial” in several places, to denote what appears to be progeny (I think). It may be best to define this in their context as it is not usual terminology that is widely known.

Response: Thanks! As suggested, we have changed it to “related varieties”.

Response to the comments of Reviewer #3:

This manuscript describes the genetic mapping and characterization of a locus, qWS8/*ipa1-2D*, that substantially contributes to traits important for yield enhancement in rice. By itself, this is an important contribution to ongoing efforts to breed and understand loci that are necessary for increasing yield in this agronomically important plant. Although I am not a breeder, this part of the manuscript seems quite well done, and clearly demonstrates that it is a non-coding region upstream of *IPA1* that causes differences in expression of this gene. This is in line with many other observations that suggest that regulatory changes are important sources of variation in traits. The authors demonstrate that the genetic change in *ipa1-2D* results in increased expression of *IPA1*, and that this expression change is associated with changes in important aspects of plant architecture. Interestingly, although qWS8 was mapped using SNPs, the key change appears to be a tandem duplication several kb upstream of the *IPA* start of transcription. The authors nicely confirm this by finding that other accessions of rice that have the SNPs but lack the tandem duplication do not exhibit changes in plant architecture. The authors demonstrate that the tandem duplication by itself, when transformed in trans, does not enhance expression of *IPA1*, nor does it, by itself, act to enhance expression of *GUS*. Although one could quibble with negative results, they were reasonable tests of the hypotheses. Surprisingly, the authors find evidence for epigenetic differences, or at least differences in DNA methylation, between NILs carrying *ipa1-2D* and those that lack it. Although fascinating, this is where I begin to have some problems with the analysis. As I will detail below, I had some technical problems with the analysis as presented. In addition however, I have some difficulties with cause effect relationships, primarily because there is no indication presented as to what actually cause the differences in DNA methylation observed here are achieved. However, as discussed below, I suspect that differential methylation region observed just upstream of the transcriptional start site of *IPA1* is in the edge of a repetitive element. This boundary effect has often been observed in maize, and is associated with differences in the chromatin environment in transposable elements immediately adjacent to genes. In any event, technical issues aside, and aside from a lack of a

final understanding of the relationship between DNA methylation, the tandem duplication and increased expression of *IPA1*, I am cautiously positive about the phenomenology presented here.

Response: Thank you for the comments and suggestion. We agree that DNA methylation might not be the direct cause of *IPA1* up-regulation, but we think the methylation status can provide clue for the possible mechanism of the *ipa1-2D*-mediated *IPA1* regulation. Therefore, as suggested we have modified the statement to make it more acceptable. We also agree that the repetitive elements upstream of *IPA1* should have impact on *IPA1* expression. As showed in the text, we found the upper region of *IPA1* is full of repeat and TE elements in the NIP genome, including the region adjacent to the transcriptional start site of *IPA1* and this might create a heterochromatin environment that depresses *IPA1* expression. This is consistent with the boundary effect found in maize, and we also found that the DNA methylation in the *IPA1* promoter can be alleviated by the *ipa1-2D* locus, possibly due to an open chromatin structure shaped by the large tandem duplication. As you suggested, we discussed the possible mechanism linking the tandem duplication, DNA methylation, and increased *IPA1* expression in the revised manuscript.

Specific comments:

line 78: “reveals a previously unidentified large tandem repeats which attenuates the epigenetic repression of *IPA1* through modifying the epigenetic landscape of the *IPA1* promoter.” Some caution is required here. Certainly, the presence of the tandem repeat is associated with increased expression, which is associated with differences in DNA methylation, but it would also seem possible that the observed changes in DNA methylation are a consequence, not a cause, of increased expression.

Response: Many thanks for the comments. Although we believe the epigenetic mechanism is involved in the *IPA1* regulation, we agree that the present data could only associate the epigenetic mark with *IPA1* expression and it would be ill-considered to state it as the direct cause of increased expression, we therefore have softened the statement in the revised manuscript.

line 81: “The superior performance of this novel *qWS8/ipa1-2D* locus will enable geneticists and breeders to molecularly design new super rice varieties with high yield.” perhaps, but if this really is an epigenetic phenomenon, this might be more difficult that the authors suspect.

Response: We agree that some epigenetic phenomenon is reversible or metastable. However, under some circumstances, epigenetic loci may be genetically stable, as recently reported in maize in which DNA methylation inheritance is stable across generations even in segregating populations (Li, Q. et al., *Genetics* 196:667-76, 2014). Consistent with it, we found that the *qWS8/ipa1-2D* can be stably inherited because the NILs have stable traits after propagation for many generations/years and we have applied this locus in our stable breeding varieties, such as JYZK-4. Nevertheless, we have modified some notions about the *qWS8/ipa1-2D*-mediated epigenetic phenomenon.

line 113: “However, Southern blotting revealed that YYP1 contains a large sequence insertion at the mapping region” I’ve done quite a bit of restriction mapping, and I have a hard time understanding how the provided Southern blot data demonstrates this, unless a corresponding map of the NIP sequence is provided, along with the expected sizes on that map as well. That being said, since the corresponding region has been sequenced in both accession, I’m not sure what the Southern data adds. Perhaps it would be better as supplemental data. Further, here and throughout the images have been heavily processed, to the point where the background cannot be seen. Given recent controversies, it would be best if all images were processed as little as possible. In figure 2b, I was a bit confused, since based on my analysis, Os08g39890.1 is IPA1, not adjacent to it. Figure 2c is far too processed, although I accept the results of the analysis.

Response: Thank you for the comment and suggestion. We are sorry to cause this confusion. The *qWS8/ipa1-2D* locus is located in the region with the triple tandem repeats. The Southern blot was performed as a direct genetic evidence to clarify the repeats in YYP1 and the single one in NIP. As suggested, we have added both the YYP1 and NIP genomic information in the schematic representation of restriction map, and moved Figure 2a to the Supplementary as new Supplementary Figure 5. We also modified the figure legend for a clearer statement. As for the background of the Southern blots, as suggested we generated new figures from raw images to replace the old ones. The predicted gene with small coding region between the repeats and *IPA1* should be *Os08g39900* but not *Os08g39890 (IPA1)*, and we have corrected this mistake in the revised manuscript. Thank you again for pointing out this.

line 116: “We sequenced a BAC containing the qWS8 locus and discovered that the qWS8 sequence” Did you also sequence the NIP region?

Responses: Yes, we also sequenced the NIP region to find all the sequence polymorphisms between two parents, and we have modified the notion to make it clear in the revised manuscript.

line 140: “We used these primers to genotype a collection of IPA varieties with varying degrees of IPA traits at two growth locations.” I’m not sure what is being portrayed here. Are these multiple accessions plotted out?

Responses: Sorry for the confusion. IPA traits have been chased in recent rice breeding programs. This experiment was performed based on genotyping and phenotyping in two growth locations to examine the *qWS8/ipa1-2D* allele in a large collection of varieties showing IPA traits, which were developed by different breeders. These accessions were analyzed to determine the *qWS8/ipa1-2D* allele distribution and its genetic contribution to the IPA traits. Accordingly we revised the sentence in the revised manuscript.

line 153: “We detected higher expression of IPA1 in plants with the YYP1 allele of qWS8 compared to plants with the NIP allele (Supplementary Fig. 7).” Since this is a quantitative argument, real time PCR should have been used, rather than regular PCR with an unknown number of cycles.

Response: Thanks. It is the semi-quantitative RT-PCR result, and we have added the

information of cycle number in the figure legend as suggested. It could clearly clarify the differential expression of the two *IPA1* alleles with such cycle number.

line 197: “We found that the region in NIL_{ipa1-2D} was more susceptible to digestion with the methyl-sensitive restriction enzymes HpaII and MspI (Fig. 4b). A similar result was obtained with other methylation sensitive enzymes (Supplementary Fig. 13).” This is where I get less comfortable. While I agree that there is some evidence for a decrease in methylation, this is restricted to only a few restriction sites. Further, the blots reveal that the effect is relatively minor, and only one sample of DNA was used for each genotype, and we have no much variation there is between samples. Finally, the XbaI/Alw44I digest is so heavily manipulated that it is hard to know what we are seeing, and there is no indication on the map which sites are differentially methylated. Although I’m tentatively convinced that there is some degree of difference in methylation in this upstream region, this analysis would have to be redone to be convincing. Repetitive sequences can be bisulfite sequences if at least one primer is in a single copy sequence. I would suggest the authors use their methyl sensitive enzyme assays to focus on specific regions that they think are hypomethylated and examine them in more detail.

Response: Thank you for the comments and suggestions. We agree that methylation-sensitive Southern blot assays can only detect a few sites, the result in our analysis directly reflect the methylation difference at several enzyme sites in the promoter region between two alleles. We have modified some statement to make it accurate and acceptable. We agree that the difference detected by Southern blotting is relatively minor because the differential methylation is quantitative and might be restricted to specific cells. However, we could repeatedly detect the difference using independent genomic DNA samples from plants grown in another season, we attach the independent assay below for your reference. We replaced the blot for XbaI/Alw44I with a better image in Supplementary Fig. 14b and modified the figure legend to make it clear. Consistent with the result of Southern blot, we have also performed bisulfite sequencing of the promoter regions and clearly showed the decreased methylation of the *qWS8/ipa1-2D* promoter (Fig. 4C-F).

Review only

Southern blot analysis of DNA methylation showing difference between NIL^{ipa1-2D} and NIL^{IPA1}. The bands reflecting different methylation pattern are denoted by arrows.

Line 198: “The blotting assays indicated the existence of highly suppressed chromatin in the IPA1 promoter that could be alleviated by ipa1-2D.” No, it simply suggests that some fraction of methylation is lost at a few sites in regions upstream of the IPA1 gene.

Response: We have changed the sentence as suggested.

line 203: “Importantly, we found a remarkable loss of cytosine methylation in the junction of two regions of NILipa1-2D compared to NILIPA1. The CHH was particularly hypomethylated in NILipa1-2D compared to NILIPA1 (Fig. 4d-f), suggesting a role for CHH demethylation in ipa1-2D-mediated up-regulation of IPA1.” This is an interesting result. A quick look suggests to me that the methylated region upstream of IPA1 is repetitive, and likely a transposon. The pattern of methylation at the margins between transposons and genes in grasses are often composed of CHH islands such as the one observed here. This would be worth more consideration. Here and in the previous blotting section, I’d like to know more about the nature of the sequences being differentially methylated.

Response: Many thanks and this is also a very interesting issue to us. Indeed, we have shown that the upstream region of *IPA1* is richened with repeat elements/transposons (Supplementary Fig. 13). The details of the region can be further found in the online database (http://rice.plantbiology.msu.edu/cgi-bin/gbrowse/rice/?name=LOC_Os08g39890). We agree that the differential CHH methylation in the *IPA1* promoter is most likely attributed to transposons/repeats’ silencing and flanking genes’ expression, and we have modified the sentence as suggested.

line 230: “As copy number increased, the tiller number decreased proportionally, whereas both stem diameter and panicle primary branch number increased” It also looks as if yield is decreasing. Is that true?

Response: Yes, we actually found that the total grain number began to decrease in transgenic lines with 2-4 copies of the transgene because overexpression of *IPA1* in the transgenic plants greatly decreased tiller numbers.

line 314-320. I would suggest that these claims are premature. The authors certainly see an association between some differences in methylation, but I don’t think they have established that it is causal. It is possible, for instance, that regulatory sequences enhance an open chromatin configuration in ipa1-2D, which, in turn results in decreases in methylation of repetitive sequences near the proximal promoters. In this case, changes in methylation would be a consequence, not a cause, of enhanced activity.

Response: Thank you for the in-depth comment. As we have responded above, we have changed the sentence as suggested.

REVIEWERS' COMMENTS:

Reviewer #3 (Remarks to the Author):

While I could still quibble about some details of this manuscript, overall, I think the authors have done a satisfactory job at addressing my previous concerns. I very much look forward to seeing the next chapters in this story.

REVIEWERS' COMMENTS:

Reviewer #3 (Remarks to the Author):

While I could still quibble about some details of this manuscript, overall, I think the authors have done a satisfactory job at addressing my previous concerns. I very much look forward to seeing the next chapters in this story.

Response: Thanks. We also look forward for the next chapters and we will do our best.